# Combined Treatment of Persistent Diabetic Macular Edema with Aflibercept and Triamcinolone Acetonide in Pseudophakic Eyes

**DOI:** 10.3390/medicina59050982

**Published:** 2023-05-19

**Authors:** Nenad Petrovic, Dusan Todorovic, Tatjana Sarenac Vulovic, Suncica Sreckovic, Fatima Zivic, Dijana Risimic

**Affiliations:** 1Clinic of Ophthalmology, University Clinical Center Kragujevac, 34000 Kragujevac, Serbia; nenadpet@yahoo.com (N.P.); tvoja.tanja@yahoo.com (T.S.V.); sunce.sun@yahoo.com (S.S.); 2Faculty of Medical Sciences, Department of Ophthalmology, University of Kragujevac, 34000 Kragujevac, Serbia; 3Faculty of Engineering, University of Kragujevac, 34000 Kragujevac, Serbia; zivic@kg.ac.rs; 4Clinic for Eye Diseases, University Clinical Center of Serbia, 34000 Kragujevac, Serbia; risimic@gmail.com; 5Medical Faculty, University of Belgrade, 11000 Beograd, Serbia

**Keywords:** diabetic macular edema, anti-VEGF, triamcinolone acetonide, pseudophakia

## Abstract

*Background and Objectives*: The main cause of the vision loss in diabetics is the development of diabetic macular edema, regardless of the stage of diabetic retinopathy. The paper aimed to examine whether the additional intravitreal application of triamcinolone acetonide to continuous anti-vascular endothelial growth factor therapy could improve therapeutic outcomes for pseudophakic eyes with persistent diabetic macular edema. *Materials and Methods*: twenty-four pseudophakic eyes with refractory diabetic macular edema, that had appeared despite three previously administered intravitreal injections of aflibercept, were divided into two groups (twelve eyes in each group). The first group continued to have aflibercept administered according to a fixed dosing regimen (once in two months). Triamcinolone acetonide 10 mg/0.1 mL (administered once per four months) was included for the second group, i.e., their treatment continued with a combination of aflibercept + triamcinolone acetonide. *Results*: The reduction in central macular thickness was higher in the eyes treated with combined therapy (aflibercept + triamcinolone acetonide) compared with the use of aflibercept alone during the entire 12-month follow-up period (3rd month *p* = 0.019; 6th month *p* = 0.023; 9th month *p* = 0.027; 12th month *p* = 0.031). As was evident from the *p*-values, the differences were statistically significant. No statistically significant difference was recorded for visual acuity: 3rd month *p* = 0.423; 6th month *p* = 0.392; 9th month *p* = 0.413; 12th month *p* = 0.418. *Conclusions*: Combined anti-vascular endothelial growth factor and steroid therapy leads to a better anatomical outcome of persistent diabetic macular edema in pseudophakic eyes, but does not lead to a more significant improvement in visual acuity than continuous anti-VEGF therapy alone.

## 1. Introduction

The main cause of vision loss in diabetics, regardless of the stage of diabetic retinopathy (DR), is the development of diabetic macular edema (DME).

The pathophysiological mechanisms underlying the development of DME are multifactorial. In addition to the up-regulation of several angiogenic and inflammatory cytokines, DR is also characterized by the increased concentrations of vascular endothelial growth factor (VEGF). That increase leads to the phosphorylation of tight junction proteins between endothelial cells and the subsequent breakdown of the blood–retinal barrier (BRB) that enlarges vascular permeability. Excessive intracellular and extracellular accumulation of fluid in the retinal tissue results in macular thickening [1].

The decrease in visual acuity (VA) mainly depends on the degree and the duration of central macular thickening. Therapy is aimed at reducing the retinal thickening and at preserving retinal function and vision. The functional response, i.e., VA improvement, depends on the anatomical condition of the macula, mainly its thickness and the integrity of the inner and outer retinal layers.

Initial therapy for center-involved diabetic macular edema (CI-DME) involves the intravitreal application of anti-VEGF (anti-vascular endothelial growth factor) drugs. This form of therapy has proved to be effective for the majority of treated eyes. The most common anti-VEGF drugs used today are aflibercept and ranibizumab. However, bevacizumab is also used off-label in clinical practice due to its lower cost. It has been reported that most DME patients responded favorably to each of the three anti-VEGF drugs within at least 2 years [2].

Although it has been confirmed that anti-VEGF agents are an effective treatment for diabetic macular edema, in some cases, that effect is limited. Studies have shown that DME persisted in 32–66% treatments with anti-VEGF injections for six months or longer, while VA was reduced [3]. The poor response to anti-VEGF treatment can be classified as anatomical or functional non-response. The partial or sub-optimal anatomical non-response exhibits as a 10–20% decrease in the central retinal macular thickness (CMT) after administration of three to five anti-VEGF injections. Functional non-response is defined as a VA increase for less than five ETDRS letters or one Snellen line after three to five anti-VEGF injections [4].

There are still no precise instructions on when and how to evaluate an insufficient response and when it is appropriate to consider alternative therapeutic options. When the failure of anti-VEGF therapy is finally confirmed, three options are possible: (a) using another anti-VEGF drug, (b) corticosteroid treatment instead of anti-VEGF therapy and (c) combining anti-VEGF and corticosteroid therapy [5].

Corticosteroid treatment exhibits several mechanisms in DME improvement. Corticosteroids stabilize the BRB by reducing retinal vessel permeability through constricting capillaries and by reducing the tight junction protein phosphorylation between endothelial cells. Furthermore, corticosteroids inhibit several cytokines, and the reduction in VEGF expression is the most important. Finally, there is strong evidence for the chronic inflammation involvement in the pathophysiology of DME, hence the anti-inflammatory properties of corticosteroids are beneficial for DME improvement [6,7]. For many years, intravitreal triamcinolone acetonide (IVTA) has been used to treat DME. Its slow-release crystalline formulation remains therapeutically active for 3 months after a single intravitreal injection. The therapeutic activity of 4 mg IVTA is retained for approximately 2–4 months and up to 9 months in the case of 20 mg IVTA [8].

However, due to possible side effects (such as cataracts and increased intraocular pressure), corticosteroids have been largely replaced by anti-VEGF. Despite this, IVTA is still used to treat DME by many physicians, especially in pseudophakic eyes and eyes that have low visual acuity and are resistant to anti-VEGF agents. Comparisons of the therapeutic outcomes of ranibizumab and TA in DME of pseudophakic eyes have shown that both medications can lead to an approximately identical CMT reduction and improvement in mean VA during the 2-year follow-up period [9].

Since anti-VEGF drugs and steroids have different but partially overlapping mechanisms, their simultaneous application should act on different DME pathophysiological mechanisms. Therefore, in eyes with persistent DME that have not responded to anti-VEGF treatment, adding steroids to anti-VEGF treatment might result in a better therapeutic outcome.

As there is still insufficient data in the literature to support the optimal integration of different treatment modalities, the aim of this research was to investigate new and different alternatives and treatment modalities for persistent diabetic macular edema in the context of its complex pathogenesis. In this study we tried to determine whether additional injections of 10 mg/0.1 mL triamcinolone acetonide in combination with anti-VEGF therapy provided better therapeutic outcome for pseudophakic eyes with refractory diabetic macular edema.

## 2. Materials and Methods

The study was conducted at the Clinic of Ophthalmology, University Clinical Center Kragujevac, Serbia, from 2020 to 2022. This was a prospective, observational, interventional, non-randomized follow-up study of 24 diabetic patients (i.e., 24 pseudophakic eyes) with persistent DME. All patients had been initially treated with three monthly doses of aflibercept intravitreally, but did not respond to the treatment adequately. The eyes with transparent ocular media were included in this study. Patients with glaucoma, with pre-existing ocular diseases other than DR and DME, or who had undergone previous ocular surgery (except cataract phacoemulsification) were not included in the study. Eyes with pronounced abnormalities of the vitreo–retinal interface were also excluded from the study.

Anatomical non-response is defined as a decrease in the central macular thickness (CMT) by less than 20% of the initial thickness after treatment with three anti-VEGF injections. Functional non-response is defined as an increase in baseline visual acuity of less than one Snellen line from the initial results, after the initial treatment with three anti-VEGF injections. The assessment of the response to the applied therapy was performed one month after the last injection of aflibercept by the two investigators.

The eyes were equally divided into two groups (twelve eyes each). Aflibercept therapy was used in all eyes, once in 2 months. In Group A (*n* = 12), no other drug was administered. For 50% of the test group, denoted as Group A+TA (*n* = 12), intravitreal injections of 10 mg/0.1 mL triamcinolone acetonide (TA) were additionally administered once in 4 months. The study was approved by the Medical Ethical Committee of the University Clinical Center Kragujevac. All participants were provided with explanations regarding the purpose of the study and the possible complications. All patients assigned to the combined treatment were provided with explanations regarding the off-label use of triamcinolone acetonide in DME treatment. All patients signed written consent forms to participate in the study.

Full ophthalmologic examinations were performed before each intravitreal application and during each follow-up control (after 1, 3, 6, 9 and 12 months). They included: best-corrected visual acuity measurement (Snellen eye charts), intraocular pressure (IOP) measurement (Goldmann applanation tonometer), biomicroscopy of transparent ocular media, fundus biomicroscopy (Goldmann three-mirror contact lens and Volk 78 lenses) and spectral domain optical coherence tomography (SD-OCT) scanning.

All of the eyes had persistent diffuse or cystic diabetic macular edema, confirmed and documented by fundus photography and fluorescein angiography (Carl Zeiss, Meditec, Inc., Dublin, CA, USA). OCT examination was performed before the treatment started and after 1, 3, 6, 9 and 12 months with spectral domain OCT (Optopol REVO NX 130 SD OCT, OPTOPOL Technology, Zawiercie, Poland). Central macular thickness (CMT) was measured automatically and values of ≥305 µm for males and ≥290 µm for females on OCT were considered as the reference values [10]. The following OCT biomarkers were analyzed before the treatments and after each subsequent control period: sub-retinal Fluid accumulation (SRF), intraretinal cystic spaces (IRCS), disorganization of retinal inner layers (DRIL), outer retinal layers (ORL), integrity of external limiting membrane (ELM) and ellipsoid zone of the photoreceptors (EZ). OCT biomarker analysis was performed in the central macular area (1 mm diameter), and all analyzed markers were graded based on scores ranging between 0 and 3: normal or absent (Grade 0, gr°), minimal (Grade 1, gr^1^), moderate (Grade 2, gr^2^) and severe (Grade 3, gr^3^). The OCT scans were independently evaluated by two retinal specialists.

Aflibercept 2 mg/0.05 mL (Eylea^®^; Bayer farmacevtska družba, Ljubljana, Slovenija) was applied by intravitreal injection once per 8 weeks (2 months) in all eyes in both study groups.

Intravitreal injection of TA was performed under sterile conditions. Preservative-free 40 mg/1 mL Kenalog was used (Kenalog, Bristol Myers Squibb, Athens, Greece) for the TA injections. Since 0.1 mL of the original Kenalog solution contains 4mg of triamcinolone, we applied the technique of triple sedimentation to obtain higher concentrations of TA. The process was described by Jonas et al. [11]. Since IOP can increase after TA administration, anti-glaucoma drugs were prescribed, but only if the IOP increased by more than 5 mmHg compared with the pre-injection values. The fixed combination of dorzolamide hydrochloride–timolol maleate (Cosopt^®^, MSD, Haarlem, Netherlands) was used for these purposes.

SPSS version 22 (IBM Corp., Armonk, NY, USA) was used for statistical analysis. The Kruskal–Wallis test was used for testing the changes in VA and IOP during the follow-up period. The statistical differences in retinal thickness between the two groups were evaluated by the two-tailed *t*-test. The chi-square test (χ^2^ test) was used to examine the incidence of SRF, IRCS, DRIL, and the integrity of ORL, ELM and EZ of the photoreceptors. The Mann–Whitney U test was used for the inter-group comparisons. The values of *p* < 0.05 were considered to be statistically significant.

## 3. Results

This trial included 24 pseudophakic eyes (24 diabetics) with refractory diabetic macular edema (DME). The patients had not responded to three previously administered intravitreal aflibercept injections. The eyes were equally divided into two groups (twelve eyes each). Group A continued their aflibercept therapy according to a fixed dosing regimen (one injection in two months). Group A+TA followed the same dosing regimen with aflibercept, but were additionally administered 10 mg/0.1ml triamcinolone acetonide injections every 4 months. By the first follow-up session, both groups had received six injections of aflibercept. The eyes treated with the combined therapy additionally received three TA injections.

The average patients’ age in Group A was 66.78 ± 6.82 (in a range of 52–73 years) and in Group A+TA it was 68.83 ± 5.86 (in a range of 55–72 years). All patients had DM type 2, with the average duration of diabetes in Group A being 17.42 ± 4.37 years, and in Group A+TA, it was 16.78 ± 5.86 years. The differences in the age and the duration of diabetes mellitus were not statistically significant (*p* = 0.446, *p* = 0.152). Group A comprised seven female and five males patients (58.3% vs. 41.7%), while Group A+TA included six female (50%) and six male patients (50%). The difference in gender distribution was not statistically significant (*p* = 0.919). In Group A, there were nine eyes (75.0%) with non-proliferative DR and three eyes (25.0%) with proliferative DR. In the group with the combined treatment, there were ten eyes (83.3%) with non-proliferative DR and two eyes (16.7%) with proliferative DR. No statistically significant difference was recorded for the evolutionary stage of retinopathy (*p* = 0.879). Persistent macular edema, which involves the central subfield of the macula, was confirmed in the right eye in seven (58.3%) and in the left eye in five (41.7%) patients in both groups. Both groups exhibited a satisfactory quality of glycemic control, i.e., their HbA1c values were below 8% during the entire follow-up period.

All examined eyes were pseudophakic. Cataract surgery had been performed with the phacoemulsification technique 3.23 ± 2.72 prior to this study in Group A and 2.89 ± 2.93 years prior to the study in Group A+TA. Prior to the initial treatment with three aflibercept injections, six eyes (50.0%) from Group A and seven eyes (58.3%) from Group A+TA had undergone focal laser photocoagulation of DME. Panretinal photocoagulation (PRP) had been previously performed in three eyes (25.0%) from Group A and two eyes (16.7%) from Group A+TA. Six eyes from the first (50%) group and five eyes from the second group (41.77%) had not had any previous laser treatment for DME, with their previous treatment only being three intravitreal aflibercept injections.

All eyes received six injections of aflibercept during the follow-up period. Group A+TA received their first TA injection (10 mg/0.1 mL) at 2.2 ± 1.1 weeks (range 2–3) after the first aflibercept injection, and were then administered new dosages once in four months (each eye received a total of three injections).

Table 1 shows the mean values of visual acuity, intraocular pressure, central macular thickness and the degree of expression of OCT biomarkers at baseline and after the follow-up controls after 1, 3, 6, 9 and 12 months. Figure 1 shows the mean values of visual acuity and Figure 2 shows the mean values of central macular thickness during the follow-up period.

In Group A, the mean CMT value was 452.67 ± 137.48 µm (361–696 µm), the average VA was 0.31 ± 0.16 (0.08–0.5) and the mean IOP was 13.87 ± 3.25 mmHg (13–20 mmHg) at baseline. These variables for Group A+TA were: 467.67 ± 143.53 µm (379–712 µm), 0.29 ± 0.14 (0.08–0.5) and 13.21 ± 2.94 mmHg (12–20 mmHg), respectively. The differences between the two groups were not statistically significant (*p* = 0.446, *p* = 0.427, *p* = 0.152).

During the first follow up, the mean value of CMT was 404.34 ± 105.21 µm, the mean VA was 0.38 ± 0.52 and the mean IOP was 13.16 ± 3.21 mmHg in Group A. The improvement of VA and CMT was statistically significant, but that was not true for the IOP values (*p* = 0.026, *p* = 0.029, *p* = 0.438). In Group A+TA, the mean value of CMT was 396.88 ± 113.72µm, the average VA was 0.36 ± 0.58 and the average IOP was 21.62 ± 3.54 mmHg (range 13–25). In six eyes (50%), an IOP increase of ≥5 mmHg was detected, and in three eyes (25%) the value was higher than 22 mmHg. The eyes with an elevated IOP were treated with antiglaucomatous drugs. The VA and CMT improvement were statistically significant, while no statistically significant difference was recorded for IOP (*p* = 0.025, *p* = 0.031, *p* = 0.008). The differences between the two groups were not statistically significant for VA and CMT (*p* = 0.432, *p* = 0.403). However, the difference in IOP values proved to be statistically significant (*p* = 0.014).

During the follow-up control, 3 months after the last injection, the same parameters were tested. In Group A, the mean CMT was 361.29 ± 125.88 µm, the mean VA was 0.41 ± 0.23 and the mean IOP was 13.96 ± 2.79 mmHg. The values obtained for VA and CMT differed significantly from those obtained at baseline (*p* = 0.024, *p* = 0.023). This was not the case with the IOP values (*p* = 0.451). Similar improvements were detected in Group A+TA for CMT and VA: the mean CMT was 313.13 ± 97.52 µm and the average VA was 0.43 ± 0.43 (*p* = 0.021, *p* = 0.028). However, this group also exhibited a decrease in IOP values (19.62 ± 4.24 mmHg, range 17–24 mmHg), which proved to be statistically significant (*p* = 0.013). In the case of six patients where the increase in IOP values appeared one month after the injection and who had been treated with anti-glaucomatous therapy, significant improvements were detected during the follow-up control (19.62 ± 3.24 mmHg, range 17–22 mmHg). The comparison of the two groups suggested that the differences in VA were not statistically significant, but this was not true for the CMT and IOP values (*p* = 0.423, *p* = 0.019, *p* = 0.025, respectively).

During the sixth-month follow-up period, the mean CMT value was 318.16 ± 97.23 µm, the mean VA was 0.42 ± 0.51 and the average IOP was 12.62 ± 3.58 mmHg in Group A. Compared with the initial values prior to the treatments, a significant improvement was recorded for VA and CMT, while the difference in IOP was not statistically significant (*p* = 0.021, *p* = 0.019, *p* = 0.452). In Group A+TA, the mean CMT value was 272.56 ± 68.92 µm, the average VA was 0.48 ± 0.38 and the mean IOP was 18.95 ± 4.78 mmHg. Compared with the initial values, the improvements in all three parameters were statistically significant (*p* = 0.017, *p* = 0.018, *p* = 0.015). When the two groups were compared, there were no significant differences in VA, but the differences in CMT and IOP values were statistically significant (*p* = 0.392, *p* = 0.023, *p* = 0.039).

Nine months after the last injection, the mean CMT value was 303.67 ± 134.67 µm, the mean VA was 0.44 ± 0.47 and the mean IOP was 13.62 ± 3.35 mmHg. In comparison with the initial values prior to the treatments, there were improvements in VA and CMT, but not in IOP (*p* = 0.019, *p* = 0.017, *p* = 0.437) in Group A. The results obtained for Group A+TA were: the mean CMT = 268.44 ± 69.87 µm, the mean VA = 0.47 ± 0.32 and the mean IOP = 17.87 ± 4.78 mmHg. A statistically significant improvement was recorded for VA, CMT and IOP of *p* = 0.016, *p* = 0.012 and *p* = 0.017, respectively. The comparison between the two groups indicated that there were no statistically significant differences in VA; the statistically significant differences were detected for the CMT and IOP values (*p* = 0.413, *p* = 0.027, *p* = 0.041).

Finally, 12 months after the last injection, the following values were recorded for Group A: mean CMT = 315.47 ± 121.34 µm, mean VA = 0.42 ± 0.58 and the average IOP = 13.87 ± 2.25 mmHg. Compared with the initial values, the differences in VA and CMT were statistically significant, but not for IOP (*p* = 0.018, *p* = 0.015, *p* = 0.429). In Group A+TA, the mean value of CMT was 278.78 ± 85.46 µm, the average VA was 0.45 ± 0.61 and the average IOP was 17.21 ± 3.34 mmHg. The values of *p* = 0.018, *p* = 0.011 and *p* = 0.021 show that the differences in all three variables were statistically significant (VA, CMT and IOP, respectively). Again, the differences between the two groups were not statistically significant in the case of VA, but were statistically significant in the case of CMT and IOP (*p* = 0.418, *p* = 0.031, *p* = 0.049).

Initially, the accumulation of sub-retinal fluid (SRF) in Group A was not observed in three eyes (24.99%); was mild (gr^1^) in four eyes (33.32%), moderate (gr^2^) in three eyes (24.99%) and severe (gr^3^) in two eyes (16.66%). In Group A+TA, SFR accumulation was not observed in two eyes (16.66%) (gr°), was mild (gr^1^) in three eyes (24.99%), moderate (gr^2^) in four eyes (33.32%) and severe (gr^3^) in three eyes (24.99%). There were no significant differences in the frequency and size of SRF between the two study groups (*p* = 0.921). The difference was not significant at the 1st follow-up control (*p* = 0.358), but it became significant 3 months after the start of treatment and remained significant until the end of the follow-up period (3rd month *p* = 0.047, 6th month *p* = 0.041, 9th month *p* = 0.038, and 12th month *p* = 0.033).

Initially, there were two eyes (16.66%) with no intraretinal cystic spaces (IRCS) (gr°) in Group A; mild (gr^1^) presence was recorded in two eyes (16.66%), moderate (gr^2^) presence in four eyes (33.32%) and severe (gr^3^) presence in four eyes (33.32%). In Group A+TA, one eye (8.33%) had no IRCS (gr°), while its presence was mild (gr^1^) in two eyes (16.66%), moderate (gr^2^) in five eyes (41.65%) and severe (gr^3^) in four eyes (33.32%). In terms of the frequency and size of IRCS, there were no significant differences between the two study groups (*p* = 0.973), and they were not recorded during the follow-up controls, as shown by the *p* values: 1st month: *p* = 0.754, 3rd month: *p* = 0.967, 6th month: *p* = 0.951, 9th month: *p* = 0.973, and 12th month: *p* = 0.877).

Initially, Group A had six eyes (49.98%) with no DRIL (gr°), while its presence was mild (gr^1^) in two eyes (16.66%), moderate (gr^2^) in two eyes (16.66%) and severe (gr^3^) in two eyes (16.66%). Group A+TA had five eyes (41.65%) with no DRIL (gr°), while its presence was mild (gr^1^) in three eyes (24.99%), moderate (gr^2^) in two eyes (16.66%) and severe (gr^3^) in two eyes (16.66%). At the initial stage, there was no statistically significant difference in DRIL in the two groups (*p* = 0.967). This did not change during the follow-up controls (1st month: *p* = 0.521, 3rd month: *p* = 0.951, 6th month: *p* = 0.973, 9th month: *p* = 0.951 and 12th: month *p* = 0.921).

Initially, three eyes (24.99%) had no disorganization of the outer retinal layers (ORL) or ELM and EZ alterations (ELM, EZ) at all (gr°), four eyes (33.32%) had mild levels (gr^1^), three eyes (24.99%) had moderate (gr^2^) levels and two eyes (16.66%) had severe levels (gr^3^) in Group A. Group A+TA had four eyes (33.32%) with gr°, four eyes (33.32%) with gr^1^, two eyes (16.66%) with gr^2^ and two eyes (16.66%) with gr^3^. At this stage, there were no significant differences regarding the frequency and magnitude of disorganization of ORL and the alteration of ELM and EZ between the study groups (*p* = 0.881). The statistical significance of the differences between the two groups was detected during the follow-up controls (1st month: *p* = 0.043, 3rd month: *p* = 0.033, 6th month: *p* = 0.027, 9th month: *p* = 0.031 and 12th: month *p* = 0.035).

Figure 3a shows persistent diabetic macular edema after three initial aflibercept injections with a highly increased central macular thickness (482 µm), and pronounced sub-retinal fluid accumulation and intraretinal cystic spaces. Figure 3b shows the same eye after 9 months, with significant resolution of edema in the central foveal zone (247 µm) and sub-retinal fluid and intraretinal cystic spaces after an additional four aflibercept and two Triamcinolone acetonide injections.

There were no cases of sterile or infectious endophthalmitis, intraocular hemorrhages or retinal detachment during the study.

## 4. Discussion

The main goal of DME therapy is to perform anatomical repair of the retinal tissue as quickly as possible, since functional improvement is impossible for patients with persistent chronic edema and advanced retinal damage. Currently, the initial therapy for center-involved diabetic macular edema (CI-DME) is the intravitreal application of anti-vascular endothelial growth factor (anti-VEGF) drugs.

Anti-VEGF agents have been confirmed to be effective in the most cases of diabetic macular edema. However, the responses are limited in some cases. Not all patients respond well to this therapy. In relation to anti-VEGF treatment, anatomical and functional non-response can occur [12,13].

Different OCT biomarkers can assist in predicting and evaluating the effects of implemented therapy. However, despite the progress of the OCT technology, in cases with chronic advanced DME and highly altered retinal architecture, it is difficult to assess various retinal layers. The drugs that induce a rapid anatomical recovery facilitate OCT assessment [14]. The most frequently evaluated OCT biomarker is retinal thickness in the central macular field (CMT). It is useful in assessing DME progression and severity, and in monitoring the response to the administered therapy. It is important to emphasize that visual acuity does not correlate with the central retinal thickness the same way it correlates with certain changes in the inner and outer retina, which develop as a result of persistent or chronic thickening of the macula.

In clinical practice, two scenarios can be distinguished. Functional non-response associated with good anatomic response could indicate irreversible photoreceptor loss or a greater degree of macular ischemia. On the other hand, good functional recovery can be accomplished even when anatomical response is absent or incomplete. Such cases can be defined as a partial response, i.e., VA ≥ one Snellen line or five ETDRS letters followed by the <20% decrease in CMT [4,15]. However, the cases of partial anatomic response (decrease in CMT between 10% and 20%) do not necessarily indicate an unfavorable final visual prognosis. Therefore, it is possible to continue therapy with the same drug or another from the same drug class.

The continuation of anti-VEGF therapy, despite initial suboptimal results, can lead to further anatomical and functional improvement of DME. In this study, we evaluated the anatomical and functional effect of treatment after three injections of aflibercept. Our results indicate that the assessment of the effect of anti-VEGF therapy should be performed only after a minimum of five to six injections. Only then can it be more reliably assessed to distinguish between response and non-response to the applied therapy. In the patients included in this study, the improvements were detected after a month and CMT had steadily decreased throughout the entire year (1st month: *p* = 0.029, 3rd month: *p* = 0.023, 6th: month *p* = 0.019, 9th month *p* = 0.017; 12th month: *p* = 0.015). At the end of the follow-up period, CMT was reduced by approximately 43.5% during one year (315.47 ± 121.34 µm vs. 452.67 ± 137.48 µm; *p* = 0.013). Furthermore, this anatomical improvement was accompanied by a functional increase in VA. In these patients, already after the 1st month and until the end of the 12th month, anti-VEGF therapy alone increased visual acuity significantly (1st month: *p* = 0.029, 3rd month: *p* = 0.023, 6th month: *p* = 0.019; 9th month: *p* = 0.017, 12th month: *p* = 0.015). Visual acuity was approximately 35.5% higher at the end of the follow-up period compared with the initial value (0.42 ± 0.58 vs. 0.31 ± 0.16; *p* = 0.011).

There are several mechanisms explaining how steroids may improve DME. The increased VEGF level is not the only factor responsible for DME development. Other VEGF-independent mechanisms are also involved in DME pathophysiology. The alterations in the regulation of several angiogenic and inflammatory cytokines lead to BRB breakdown. The balance between them can change as DME develops. Vascular dysfunction is dominant during the earlier DME stages, while chronic inflammatory mechanisms are more pronounced in the advanced stages of chronic DME. In clinical practice, it is very difficult to determine which dysfunction is dominant at a given moment; thus, it is hard to select the optimal DME treatment [16,17].

Currently, corticosteroids are used as the second-line treatment for DME in eyes that respond sub-optimally to anti-VEGF drugs [18,19]. There is no consensus regarding the indications for intravitreal corticosteroids. Currently, physicians have three corticosteroid options for intravitreal application: triamcinolone acetonide, dexamethasone implant (DEX) and fluocinolone acetonide insert. They are different from each other in their lipophilicity, binding affinities for glucocorticoid receptors and proteins that they regulate [6].

The DRCR.net Protocol I RCT compared ranibizumab with prompt or deferred photocoagulation with triamcinolone acetonide with prompt photocoagulation. The results showed that CMT reduction in phakic eyes was approximately similar in both groups after a 2-year follow-up period. However, visual acuity was significantly greater in the ranibizumab group. However, in pseudophakic eyes with DME, triamcinolone acetonide produced a similar improvement in mean VA to ranibizumab. This indicates that in the phakic eyes, lower visual acuity results from cataracts; however, in pseudophakic eyes, TA may be equally effective as ranibizumab in reducing retinal thickening and in improving VA (but with an increased risk of IOP elevation). In addition, in pseudophakic eyes, corticosteroids are also more appropriate due to the less frequent need for treatment, and also in cases of patients with severe cardiovascular diseases. In these cases, the use of steroids may be considered as a first-line therapy for DME [9].

Since the functional response to corticosteroids decreases with longer durations of chronic DME, the timely introduction of corticosteroid therapy is needed so that irreversible vision loss can be prevented [20].

Knowing that anti-VEGF drugs and steroids have different, but partially overlapping mechanisms, it is logical that their simultaneous application can act on different pathophysiological mechanisms responsible for DME development. In cases of persistent DME that are unresponsive to anti-VEGF treatment, the addition of steroid-based therapy can lead to better therapeutic outcomes. The rational argument for combined treatments is based on the fact that corticosteroids inhibit several cytokines including VEGF. Hence, the combination of these treatments can target all factors responsible for DME development. The results of many studies have confirmed that this type of combined therapy results in a higher decrease in the retinal thickness than anti-VEGF therapy alone. However, anatomical improvements are not always accompanied by the improvements in visual acuity. There is a discrepancy between the anatomical and functional results of combined therapy. Kim et al. [18] evaluated the short-term efficiency of TA in bevacizumab-resistant DME and noted a significant reduction in CMT at one month and three months after TA injection; however, VA improvement was recorded only after one month, but not after the third month.

Our findings indicate that combined aflibercept + TA therapy already reduced CMT after one month, and the trend remained the same during the entire follow-up period (1st month: *p* = 0,031, 3rd month: *p* = 0.028, 6th month: *p* = 0.018, 9th month: *p* = 0.012, 12th month: *p* = 0.011). At the end of the 12-month follow-up period, CMT was approximately 67.7% lower than the initial values (278.78 ± 85.46 µm vs. 467.67 ± 143.53 µm; *p* = 0.007). This anatomical improvement was accompanied by a functional increase in VA. VA was significantly higher compared with the initial values after one month and throughout an entire year (1st month: *p* = 0,025, 3rd month: *p* = 0.021, 6th month: *p* = 0.017, 9th month: *p* = 0.016, 12th month: *p* = 0.018). After a year, VA was approximately 55.2% higher than the initial values (0.45 ± 0.61 vs. 0.29 ± 0.14; *p* = 0.011).

The results of our study indicate that that both types of treatment (aflibercept alone or aflibercept + TA) can lead to successful CMT reduction and VA improvement. However, the inter-group analysis indicated that, starting from the 3-month follow-up to the 12-month follow-up, the CMT reduction was significantly higher in the eyes treated with the combined aflibercept + TA therapy than in the eyes treated only with aflibercept (1st month: *p* = 0.403, 3rd month: *p* = 0.019, 6th month: *p* = 0.023, 9th month: *p* = 0.027, 12th month: *p* = 0.031). Anatomical improvements were not accompanied by proportional functional improvements. During the same time period (from the 3rd to the 12th month), VA improvements were higher in the eyes treated with aflibercept + TA than in those treated only with aflibercept, but these differences were not statistically significant (1st month: *p*= 0.432, 3rd month: *p* = 0.423, 6th month: *p* = 0.392, 9th month: *p* = 0.413, 12th month: *p* = 0.418).

The DRCR.net conducted a clinical, randomized controlled trial to compare the effects of combined treatment (ranibizumab and dexamethasone implant) with prolonged ranibizumab treatment of refractory DME in patients who had already received three anti-VEGF injections. After 6 months, the anatomical improvements were evident in both groups, but the combined treatment was more superior (100 µm vs. 62 µm). On the other hand, no significant difference was recorded for VA. It is important to note that this study included both phakic and pseudophakic eyes. Their subgroup analysis revealed that pseudophakic eyes had a better visual acuity outcome with combined treatment [21].

In this paper we investigated the effect of the combined use of anti-VEGF drugs and steroids, and we used aflibercept and triamcinolone acetonide for several reasons. We applied a concentrated dose of 10 mg/0.1 mL of TA, which remains therapeutically active in the eye for approximately 4 months [8]. Many authors have suggested that the clinical efficacy of an intravitreal dexamethasone implant is limited to 4 months in most cases, so we achieved a similar steroid effect in the eye with a concentrated dose of 10 mg/0.1 mL TA. Moreover, there is currently no established procedure for the treatment of DME with DEX implants, and the optimal interval between injections or the impact of the loading dose remain undetermined. We wanted to investigate new combined therapeutic alternative modalities to improve the treatment outcome of persistent DME. Although posterior sub-Tenon injection of triamcinolone acetonide is less invasive than its intravitreal application, it was indicated that changes in central macular thickness and visual acuity after intravitreal administration are more effective than after sub-Tenon injection for the treatment of refractory diabetic macular edema. That was the reason why the intravitreal approach was chosen [22].

Our research showed that both types of DME treatment (aflibercept only or the combination of aflibercept + TA) can be very successful in improving various OCT biomarkers. Subretinal fluid accumulation (SRF) between the sensory retina and RPE, which leads to subfoveal neurosensory detachment, is a very important OCT biomarker. It is noticed in 15–30% of patients with macular edema. It has been indicated that the presence of SRF can be associated with good anatomical and functional responses. The biochemical analysis of subretinal fluid revealed elevated levels of interleukin (IL)-6, which further indicates that inflammatory mechanisms play a significant role in these eyes. Our findings indicated that combined therapy was statistically superior in reducing SFR as compared to the use of aflibercept alone, from the 3rd month until the end of the follow-up period (1st month: 0.358, 3rd month: *p* = 0.047, 6th month: *p* = 0.041, 9th month: *p* = 0.038 and 12th month: *p* = 0.033).

The hyperpermeability of retinal vessels changes the osmotic gradient in retinal tissue, which leads to fluid accumulation in retinal interstitial tissue and the formation of intraretinal cystic spaces (IRCS). Large empty cysts (≥200 µm), if localized in the outer nuclear layer, negatively affect visual function due to a disrupted junction between the photoreceptors’ inner and outer segments (IS/OS). Thus, the presence of intraretinal cysts is an indicator of structural retinal damage and the chronicity of DME. Regarding the reductions in the number and size of intraretinal cysts, our findings indicated that there were no significant differences between the two groups during the entire follow-up period (1st month: *p* = 0.754, 3rd month: *p* = 0.967, 6th month: *p* = 0.951, 9th month: *p* = 0.973 and 12th month: *p* = 0.877).

Disorganization of the retinal inner layers (DRIL) represents the inability to visualize the border between the layers of the inner retina on OCT images, i.e., between the ganglion cells, inner plexiform, inner nuclear and outer plexiform layer. The existence of DRIL indicates the chronicity of edema. A size of >500 µm is considered to be a poor prognostic indicator for functional improvements even after the resolution of edema. Our findings indicated that there were no significant differences in DRIL repair between the two study groups (1st month: *p* = 0.521, 3rd month: *p* = 0.95, 6th month: *p* = 0.973, 9th month: *p* = 0.951 and 12th month: *p* = 0.921).

The outer retinal layers (ORL) involve the space between the ELM and RPE, which includes the rod and cone cell bodies, and inner and outer photoreceptor segments. The condition of the outer retinal layers directly reflects the integrity of the photoreceptors. As edema progresses, the outer retinal layers thicken. It has been highlighted recently that ORL thickness better correlates with VA than the total retinal thickness. The ELM and EZ integrity directly reflect the condition of photoreceptor cell bodies. ORL alterations are seen as a poor prognostic sign as many studies have indicated an incomplete and delayed visual recovery in patients with ORL disruption. Visual acuity levels are positively correlated with ELM and EZ recovery rates. Intravitreally applied bevacizumab can cause ELM repair, and then EZ restoration. In addition, there is evidence that a dexamethasone implant can effectively repair ORL ultrastructural damage and lead to better visual outcomes. When it comes to ORL, ELM and EZ repair, the combined therapy exhibited statistical superiority over the use of aflibercept alone (1st month: *p* = 0.043, 3rd month: *p* = 0.033, 6th month: *p* = 0.027, 9th month: *p* = 0.031 and 12th month: *p* = 0.035).

Our results indicated a greater superiority of the combined therapy in terms of SRF resolution and the repair of the outer retinal layers over the anti-VEGF therapy used alone. These findings suggest that steroids affect the outer retinal layers, RPE and choroid more strongly than anti-VEGF therapy alone. The abnormalities of these retinal layers do not solely result from VEGF up-regulation. The different types of inflammatory cells and cytokines have been detected in the choroid during DR. The choroidal abnormalities may be sensitive to steroids exactly because inflammation plays a significant role in DME pathophysiology [23].

The most frequent side effects of IVTA injections are IOP increase and cataract formation. It is known that triamcinolone acetonide is very lipophilic and that its crystals are deposited more in the trabeculae, thus leading to a significant increase in IOP and cataract formation. It should be noted that only pseudophakic eyes were included in our study, so we could not consider cataract development as an adverse event at all. Steroid-induced ocular hypertension was observed in one-third to more than half of the eyes which received IVTA injection [24]. In our study, IOP increases of ≥5 mmHg was detected in six eyes (50%) treated with the combined therapy during follow up after the first month. In three eyes (25%), IOP increased by over 22 mmHg with respect to the initial levels. For the eyes in which IOP growth was detected one month after the IVTA injection, anti-glaucomatous therapy was added. Three months after the IVTA injection, there was a significant IOP decrease in all six patients until the end of the follow-up period. Despite IOP normalization, the mean IOP was significantly higher during the entire follow-up period in the eyes treated with the combined therapy compared to those treated with aflibercept alone.

## 5. Conclusions

Additional triamcinolone acetonide application to continuous anti-VEGF therapy provides a better anatomical therapeutic outcome of persistent diabetic edema in pseudophakic eyes compared with anti-VEGF therapy alone. The addition of steroids to diabetic macular edema treatment has a pronounced effect on the reduction in central macular thickening, repair of the outer retinal layers and the resolution of subretinal fluid. Except for a moderate IOP increase, which can be satisfactorily controlled, intravitreal application of TA in pseudophakic eyes has an acceptable safety profile without ocular toxicity for both short- or long-term periods. The main limitation of our study was the small number of analyzed eyes to be able to draw a reliable conclusion regarding whether the additional application of 10 mg/0.1 mL of TA to the continuous administration of anti-VEGF therapy leads to a better therapeutic outcome of persistent diabetic macular edema in pseudophakic eyes. Although the improved anatomical status of retinal tissue probably enables the preservation of retinal function; however, additional steroid therapy does not lead to a more significant visual acuity improvement than continuous anti-VEGF therapy alone. Longer follow-up is needed to estimate the long-term effect of such an improved retinal anatomical condition on the preservation of macular function and visual acuity.

## Figures and Tables

**Figure 1 medicina-59-00982-f001:**
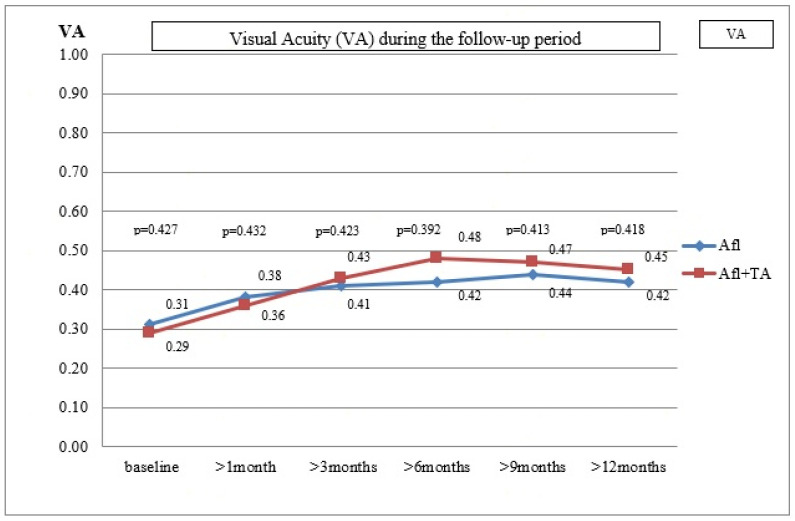
The mean values of visual acuity.

**Figure 2 medicina-59-00982-f002:**
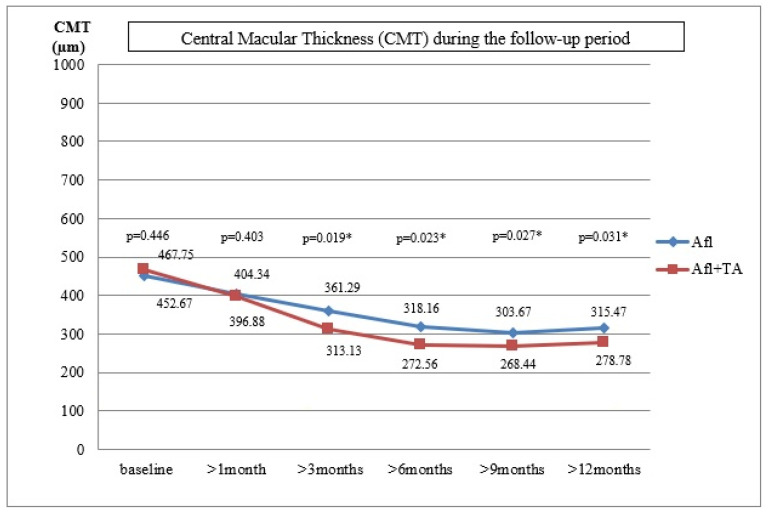
The mean values of central macular thickness during the follow-up period.

**Figure 3 medicina-59-00982-f003:**
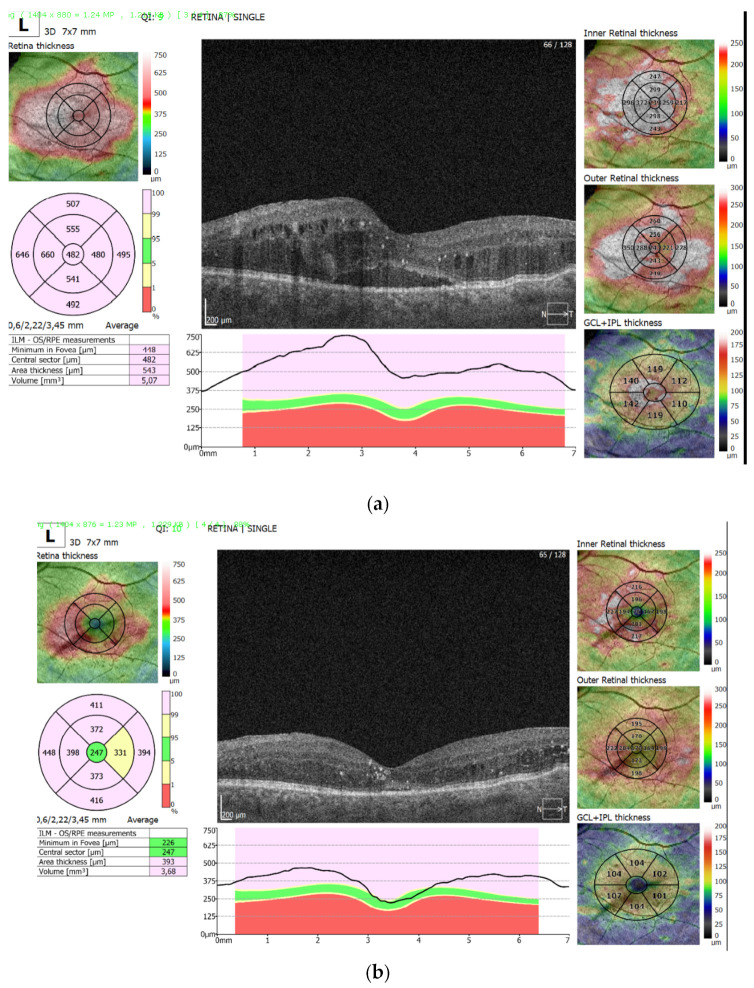
(**a**)—persistent diabetic macular edema after three initial aflibercept injections with a highly increased central macular thickness (482 µm); (**b**)—the same eye after 9 months, with significant resolution of edema in the central foveal zone (247 µm).

**Table 1 medicina-59-00982-t001:** Mean values of visual acuity, intraocular pressure, central macular thickness and the degree of expression of OCT biomarkers at baseline and after the follow-up controls after 1, 3, 6, 9 and 12 months.

		Baseline		>1 Month		>3 Months		>6 Months		>9 Months		>12 Months	
Group		A	A+TA	A	A+TA	A	A+TA	A	A+TA	A	A+TA	A	A+TA
VA (Snellen)		0.31 ± 0.16	0.29 ± 0.14	0.38 ± 0.52	0.36 ± 0.58	0.41 ± 0.23	0.43 ± 0.43	0.42 ± 0.51	0.48 ± 0.38	0.44 ± 0.47	0.47 ± 0.32	0.42 ± 0.58	0.45 ± 0.61
		*p* = 0.427		*p* = 0.486		*p* = 0.413		*p* = 0.392		*p* = 0.478		*p* = 0.418	
IOP (mmHg)		13.87 ± 3.25	13.21 ± 2.94	13.16 ± 3.21	21.62 ± 3.54	13.96 ± 2.79	19.62 ± 3.24	12.62 ± 3.58	18.95 ± 3.78	13.62 ± 3.35	17.87 ± 3.91	13.87 ± 2.25	17.21 ± 3.34
		*p* = 0.152		*p* = 0.014 *		*p* = 0.025 *		*p* = 0.039 *		*p* = 0.041 *		*p* = 0.049 *	
CMT (µm)		452.67 ± 137.48	467.67 ± 143.53	404.34 ± 105.21	396.88 ± 113.72	361.29 ± 125.88	313.13 ± 97.52	318.16 ± 97.23	272.56 ± 68.92	303.67 ± 134.67	268.44 ± 69.87	315.47 ± 121.34	278.78 ± 85.46
		*p* = 0.446		*p* = 0.403		*p* = 0.019 *		*p* = 0.023 *		*p* = 0.027 *		*p* = 0.031 *	
SRF (%)	gr°	24.99	16.66	41.65	41.65	41.65	58.31	58.31	74.97	66.64	83.3	66.64	83.3
	gr^1^	33.32	24.99	16.66	24.99	24.99	33.32	16.66	8.33	8.33	16.66	8.33	16.66
	gr^2^	24.99	33.32	33.32	24.99	24.99	8.33	16.66	16.66	16.66	0	8.33	0
	gr^3^	16.66	24.99	8.33	8.33	8.33	0	8.33	0	8.33	0	16.66	0
	*p*	*p* = 0.921		*p* = 0.358		*p* = 0.047 *		*p* = 0.041 *		*p* = 0.038*		*p* = 0.033 *	
IRCS (%)	gr°	16.66	18.33	41.65	33.32	49.98	49.98	66.64	58.31	66.64	74.97	66.64	74.97
	gr^1^	16.66	16.66	16.66	41.65	33.32	24.99	24.99	33.32	33.32	24.99	24.99	24.99
	gr^2^	33.32	41.65	16.66	16.66	16.66	16.66	8.33	8.33	0	0	8.33	0
	gr^3^	33.32	33.32	8.33	8.33	0	8.33	0	0	0	0	0	0
	*p*	*p* = 0.973		*p* = 0.754		*p* = 0.967		*p* = 0.951		*p* = 0.973		*p* = 0.877	
DRIL (%)	gr°	49.98	41.65	58.31	41.65	58.31	49.98	58.31	49.98	66.64	58.31	66.64	66.64
	gr^1^	16.66	24.99	24.99	33.32	24.99	33.32	24.99	33.32	24.99	24.99	33.32	24.99
	gr^2^	16.66	16.66	8.33	8.33	8.33	8.33	16.66	16.66	8.33	16.66	0	8.33
	gr^3^	16.66	16.66	8.33	16.66	8.33	8.33	0	0	0	0	0	0
	*p*	*p* = 0.967		*p* = 0.521		*p* = 0.951		*p* = 0.973		*p* = 0.951		*p* = 0.921	
ORL ELM EZ(%)	gr°	24.99	16.66	24.99	41.65	41.65	66.64	49.98	74.97	58.31	83.3	58.31	74.97
	gr^1^	33.32	33.32	41.65	33.32	33.32	16.66	24.99	16.6	16.66	8.33	16.66	16.6
	gr^2^	24.99	16.66	16.66	16.66	16.66	16.66	16.66	8.33	16.66	8.33	16.66	8.33
	gr^3^	16.66	16.66	16.66	8.33	8.33	0	8.33	0	8.33	0	8.33	0
	*p*	*p* = 0.881		*p* = 0.043 *		*p* = 0.033 *		*p* = 0.027 *		*p* = 0.031 *		*p* = 0.035 *	

Abbreviations: Group A—aflibercept group alone, Group A+TA—aflibercept + triamcinolone acetonide, VA—visual acuity, IOP—intraocular pressure, CMT—central macula thickness, SRF—sub-retinal fluid accumulation, IRCS—intraretinal cystic spaces, DRIL—disorganization of retinal inner layers (DRIL), ORL—outer retinal layers, ELM—external limiting membrane integrity, EZ—ellipsoid zone of the photoreceptors integrity. *p*-values compared to baseline; * Statistical significance.

## Data Availability

The data presented in this study are available on request from the corresponding author. The data are not publicly available due to ethical restrictions.

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
