# Peer review of "Combined Treatment of Persistent Diabetic Macular Edema with Aflibercept and Triamcinolone Acetonide in Pseudophakic Eyes"

_medicina, 2023, doi:10.3390/medicina59050982_

Round 1

Reviewer 1 Report

This study by Nenad et al. proposes the notion that combined treatment with aflibercept and triamcinolone acetonide can be effective in persistent diabetic macular edema. They also reviewed clinical trials about the treatment strategies for DME and multiple biomarkers changes such as DRIL or ORL disruption.

The manuscript is interesting and generally well-written, and the presentation of the content in Tables/Figures is good. However, I have some concerns the authors should take into account (see detailed suggestions below).

1.      The authors stated that they included patients who did not respond to the treatment adequately. However, figure 1 shows that the Afl group did show vision improvement and CMT reduction after repetitive injections. Were the study participants really ‘nonresponders?’ Basement demographic data comparing both group participants should be presented.

2.      They compared the anatomical/functional outcome between the eyes only treated with aflibercept and those treated with both aflibercept and triamcinolone. It is not surprising that the eyes treated more present a better outcome. A combination of additional triamcinolone can be considered when the additional treatment did not cause any serious adverse effect threatening vision, including cataract development, IOP elevation, and infectious endophthalmitis. The number of injection and occurrence rate of side effects should be presented in a table.

3.      Some eyes need to be treated by combining corticosteroids with anti-VEGF to achieve DME resolution, as they stated. If the eyes need to be treated with steroids constantly, there are good treatment choices for dexamethasone suspension, such as Ozurdex. Why do the authors give repetitive injections with triamcinolone acetonide? Do they want to know a difference between triamcinolone and dexamethasone?

4.      The manuscript is generally hard-to-read because it is too long and does not focus on the facts they want to enlighten in the research. Discussion needs to be shortened. For example, a review of DRCR.net protocol T with a comparison of different anti-VEGF agents is not a necessary discussion for this study because this study only adopted aflibercept as an anti-VEGF drug.   

5.      Periods (.) and commas (,) are used inconsistently in Table 1 and Figure 1. Some numbers are inadequately written (e.g., 0.403 - 403)

6.      Error bars should be present in Figure 1.

7.      The manuscript should be reviewed for typographical errors. (e.g., called – called in 4th line, 3rd paragraph on page 10)

The manuscript should undergo a thorough English editing. 

Author Response

Dear reviewer,

thank you for your revision of our manuscript titled: “Combined treatment of persistent diabetic macular edema with aflibercept and triamcinolone acetonide in pseudophakic eyes”.  We made all the corrections as you have suggested.

Best regards

Dr Dusan Todorovic

Reviewer 2 Report

The authors employed aflibercept and/or intravitreal triamcinolone injection for patients with diabetic macular edema refractory to prior 3 injections of aflibercept. 

The authors used both IS/OS and ellipsoid zone (EZ) in this manuscript. Please use solely the latter throughout the manuscript.

What did not authors use sub-tenon injection of triamcinolone, and why did not they compare? Also please discuss the difference of clinical outcome.  

In discussion, the description is too long to lose what the important points are. Please remove the description overlapping between introduction and discussion.

Author Response

(The authors gave the same response as above.)

Round 2

Reviewer 1 Report

Suggested questions were answered appropriately.